# Improved Activity of Herbal Medicines through Nanotechnology

**DOI:** 10.3390/nano12224073

**Published:** 2022-11-18

**Authors:** Mayang Kusuma Dewi, Anis Yohana Chaerunisaa, Muhaimin Muhaimin, I Made Joni

**Affiliations:** 1Doctoral Study Program, Faculty of Pharmacy, Universitas Padjadjaran, Jl. Raya Jatinangor Km 21,5, Sumedang 45363, Indonesia; 2Department of Pharmaceutics and Pharmaceutical Technology, Faculty of Pharmacy, Universitas Padjadjaran, Sumedang 45363, Indonesia; 3Department of Pharmaceutical Biology, Faculty of Pharmacy, Universitas Padjadjaran, Sumedang 45363, Indonesia; 4Functional Nano Powder University Center of Excellence (FiNder U CoE), Universitas Padjadjaran, Jalan Raya Bandung-Sumedang Km 21, Jatinangor 45363, Indonesia; 5Department of Physics, Faculty of Mathematics and Natural Science, Universitas Padjadjaran, Jl. Raya Bandung-Sumedang Km 21, Jatinangor 45363, Indonesia

**Keywords:** phytochemical, herbal medicine, drug delivery, nanotechnology, bioactivity

## Abstract

Phytochemicals or secondary metabolites are substances produced by plants that have been shown to have many biological activities, providing a scientific basis for using herbs in traditional medicine. In addition, the use of herbs is considered to be safe and more economical compared to synthetic medicine. However, herbal medicines have disadvantages, such as having low solubility, stability, and bioavailability. Some of them can undergo physical and chemical degradation, which reduces their pharmacological activity. In recent decades, nanotechnology-based herbal drug formulations have attracted attention due to their enhanced activity and potential for overcoming the problems associated with herbal medicine. Approaches using nanotechnology-based delivery systems that are biocompatible, biodegradable, and based on lipids, polymers, or nanoemulsions can increase the solubility, stability, bioavailability, and pharmacological activity of herbals. This review article aims to provide an overview of the latest advances in the development of nanotechnology-based herbal drug formulations for increased activity, as well as a summary of the challenges these delivery systems for herbal medicines face.

## 1. Introduction

The traditional use of plants, both used directly and extracted, for medicinal purposes has existed since ancient times [1,2]. Plants are a source of various phytochemicals and have been used for human health because of their low side effects, low cost, and high acceptance by the general public [3].

Phytochemicals are substances produced by plants (also known as secondary metabolites) that play an important role in traditional medicine. Secondary metabolites have been shown to exhibit various biological activities, which provide a scientific basis for the use of herbs in traditional medicine. They show pharmacological effects that can be used to treat bacterial and fungal infections and even chronic degenerative diseases, such as diabetes and cancer [4,5].

Herbal medicines are increasingly popular throughout the world and have promising potential to provide treatment, maintain and improve health, as well as prevent and treat several diseases because they are considered safe compared to modern conventional medicines and are more economical [6,7]. However, most of these biologically active phytochemical constituents have limitations; namely, their absorption and distribution are low, and the target specificity of phytochemicals is generally low, which results in low bioavailability, resulting in decreased biological activity. In addition, large doses are required to produce the activity of these phytochemical compounds, and also some of these phytochemical compounds are sensitive to acidic conditions and have low stability phytochemical compounds [6,8,9,10,11]. These limits hinder their clinical application.

Puerarin is the main active compound isolated from the roots of *Pueraria lobata* (Willd.) Ohwi, which has broad pharmacological activity. Puerarin is used for cardiovascular disease, diabetes, osteonecrosis, Parkinson’s disease, Alzheimer’s disease, endometriosis, and cancer [12]. Puerarin has a low solubility in water: 0.46 mg/mL. The maximum solubility of puerarin occurs in a phosphate buffer of pH 7.4 at 7.56 mg/mL [13,14]. The low solubility limits the application of puerarin. In recent years, research on increasing the bioavailability of puerarin has grown rapidly. Various nanotechnology has been investigated to increase the bioavailability of puerarin, one of which is the solid lipid nanoparticle (SLN) carrier system. Compared to puerarin suspension, SLN-puerarin is absorbed rapidly. This is supported by a shorter Tmax. In addition, SLN-puerarin showed more than threefold bioavailability compared to puerarin suspension [15,16].

Nanotechnology-based delivery systems function as drug carriers that can overcome the various limitations that herbal medicines face, including increasing the bioavailability and bioactivity of phytochemicals. The approach using nanotechnology can be a promising innovative technology that is applied to phytochemical constituents, increasing the phytotherapy efficiency of herbal medicines. The development of an efficient and safe drug delivery system is the goal of various researchers. Recent developments in the field of nanotechnology have led to renewed interest in herbal medicinal formulations. Several delivery system approaches, such as phytosomes, solid lipid nanoparticles (SLN), nanostructured lipid carriers (NLC), polymeric nanoparticles, nanoemulsions, etc., have been proposed. Nanoparticles have been used to modify and improve the pharmacokinetic properties of different drugs, so the nanotechnology approach is expected to increase the bioavailability and bioactivity of herbal medicines [6,10,17,18]. This article review aims to provide an overview of the latest advances in the development of nanotechnology-based herbal drug formulations that increases herbal activity.

## 2. Nanotechnology

The nanoscale system has a particle diameter of 0.1 μm, otherwise known as a submicrometer. This provides several advantages with regard to various aspects, including the route of administration and increased therapeutic effects, which makes this nanotechnology more developed and widely studied by researchers. Many studies have combined herbal medicine with nanotechnology because nano-sized systems can increase activity, reduce dosages, and minimize side effects. 

Herbal medicines using nanotechnology-based delivery systems have great potential and unique properties, such as being able to convert less soluble, poorly absorbed, unstable substances into promising drugs. Therefore, nanotechnology-based delivery systems represent a promising prospect for enhancing herbal activity and overcoming the dilemmas associated with herbal medicine (Figure 1) [19].

## 3. Nanotechnology-Based Drug Delivery System for Phytochemical Compounds

According to the literature, 70% of the active ingredients obtained from plants are hydrophobic [3]. New technology has been used as a strategy to increase the bioavailability/bioactivity of phytochemical compounds. In order to develop new nanotechnology-based therapies, the ability to design suitable formulations for drug delivery is of the utmost importance. Phytochemical delivery is essential for effective disease prevention and treatment. These delivery systems include lipid-based delivery systems and polymer-based delivery systems, which have the potential to increase the bioactivity of phytochemical compounds (Figure 2) [22,24,25].

New drug delivery systems based on nanotechnology have been developed for effective herbal drug delivery [27]. Lipid-based carrier systems (Figure 3) consisting of vesicular systems (liposomes, phytosomes, transfersomes, ethosomes, and niosomes), lipid particulates (SLN and NLC), and nanoemulsions have garnered particular interest for phytochemical delivery aimed at increasing bioactivity and bioavailability, as well as the stability of phytochemical compounds [3].

Vesicular drug delivery systems can be defined as highly ordered assemblies consisting of one or more concentric bilayers formed as a result of self-assembly in the presence of water [28]. SLN and NLC are two types of nanoparticle systems consisting of lipid cores formed from solid lipids or mixtures with liquid lipids [29]. Nanoemulsions are used to increase the bioavailability of hydrophobic drugs and drugs with a high first-pass metabolism. The components of the nanoemulsion system include oils, lipids, surfactants, water-soluble cosolvents, and water [30].

Lipid-based nano systems represent the largest and most investigated nanocarrier category. Typically, these carriers exhibit a lower toxicity profile and a more reasonable cost compared to polymer carriers [31]. Among all approaches, the delivery of liposomes and phytosomes is considered to be very effective [32]. With high biocompatibility and biodegradability, liposomes offer the ability to improve the solubility, efficacy, and bioavailability of drugs. Liposomes can be used to encapsulate hydrophilic and lipophilic drugs [33,34].

Curcumin and tetrandrine are phytochemical compounds that have poor solubility; therefore, their bioavailability is low [35,36]. To overcome this problem, Song, et al. (2022) encapsulated curcumin and tetrandrine in a liposome carrier system [37]. The results showed high solubility and bioavailability for curcumin and tetrandrine after their encapsulation within the liposome carriers. In addition, in vivo studies have shown no significant toxicity of this system to Zebrafish [37]. However, liposomes have stability issues and experience self-leakage with particular drugs. Therefore, to protect the stability of liposomes, the stabilizer DSPE–MPEG 2000 (DP) has been widely used [31]. The addition of DP not only significantly increases the particle size distribution but also significantly increases the encapsulation efficiency and loading dose of curcumin and tetrandrine [38].

Lipids are considered safe and useful materials for drug delivery. The stability of the system is closely related to the distribution of particle size, lipid content, and surfactant content, which are able to stabilize the dispersion [37].

Polymers can be used as phytochemical carriers to form structures, such as dendrimers, micelles, polymersomes, etc. The precise classification of polymeric nanocarriers needs further study because, in biomedical and pharmaceutical nanotechnology, no specific limits have been defined. Therefore, there may exist differing perceptions. However, a brief sketch of the polymer-based nanocarrier is more or less depicted in Figure 4. [24]. Examples of natural and synthetic polymers that are often used are alginate, albumin, chitosan, starch, polylactic acid, and Poly(glycolide-co-lactide) (PGLA) [39,40]. Polymers have played an important role in advancing drug delivery technology by providing the controlled release of the active substance in a constant dose over a long period of time [41]. Polymers protect the drugs via encapsulating, entrapping (inside the core), conjugating, or adsorbing them onto the particle surface [42] Generally, natural polymers are nontoxic, biocompatible, and biodegradable. Polysaccharides are polymers that are commonly used for the manufacture of polymer-based nanoparticles [43].

In this section, we will discuss one of the lipid-based drug delivery systems, namely, phytosomes, that can increase the activity of phytochemical compounds. Phytosomes represent one of the delivery systems that is widely used to overcome problems regarding phytochemical compounds, such as low solubility, which causes low bioavailability so that the resulting effect is not optimal. In addition, this section discusses the application of polymer-based delivery systems that can increase the activity of phytochemical compounds (Table 1 and Table 2).

Phytosomes are vesicular delivery systems that can increase the absorption and bioavailability of active ingredients that have low solubility [47]. Phytosomes are composed of phospholipids and phytochemical constituents, which form hydrogen bonds through the reaction between the polar groups of phosphatidylcholine and the plant extracts in aprotic solvents [48]. Phytosome formulations can increase the activity of the phytochemical compounds from herbal plants. 

**Table 2 nanomaterials-12-04073-t002:** Phytosome delivery system application.

Bioactive Compound	Nanoparticle Type	Method of Preparation	Nanoparticle Characteristics *	Drug Release/Pharmacokinetic Properties	Experiment/Model/Dose/Route	Improvement Activity	Toxicity	Ref
Sinigrin	Phytosomes	Antisolvent precipitation	PS: 153 ± 39 nm	-	Dose: 0.14 mg/mL. In vitro, evaluated on HaCaT cells.	Wound-healing	Sinigrin, phytosome, and phytosome blank (without drug) showed minimal cytotoxicity to HaCaT cells at all concentrations (0.048 mg/mL–0.14 mg/mL).	[49]
			ZP: 10.09 ± 0.98mV					
			%EE: 69.5 ± 5					
Taxifolin-rich ethyl acetate fraction	Phytosomes	Antisolvent precipitation	PS: (544 nm)	-	In vitro antioxidant activity and ex vivo anticancer investigation by MTT and TB assay using MCF7 cell lines	Antioxidant	-	[50]
			ZP: −28.1					
			PDI: 0.141					
			%EE: 75.40 ± 0.53%					
Silybins	Phytosomes-nanosuspension	Antisolvent precipitation- high-pressure homogenization	PS: 223.50 ± 4.80 nm	Increased dissolution rate in vitro increased plasma concentration in vivo	In vivo: carbon tetrachloride (CCl_4_) induced hepatic injury model mice	Hepatoprotection	-	[51]
			PDI: 0.217 ± 0.011					
			ZP: −23.14 ± 2.73 mV					
Apigenin	Phytosomes	Solvent evaporation	PS: 107.08 ± 1.30 nm	36-fold increase in water solubility and increase in oral bioavailability	Dose: 25 mg/kg	Antioxidant	-	[52]
			PDI: 0.37 ± 0.012		Carbon tetrachloride-induced liver function of a rat model			
Quercetin	Phytosomes	Thin film hydration method	PS: 70 ± 7.44 nm		The estrogenic activity of Quercetin-phytosome was investigated in an ovariectomized rat model using 10 and 50 mg/kg/oral doses for 4 weeks	Hormone replacement therapy	-	[53]
			%EE: 98.4%		IC50 quercetin: 13.9 μg/mL			
			ZP: −44.6 ±		IC50 quercetin-phytosome: 11.4 μg/mL			
			4.1 mv					
Gingerol	Phytosome	Antisolvent precipitation	PS: 431.21 ± 0.90 nm	Drug release: sustained release	In vitro microbial study: agar well diffusion and dilution method. In vitro Anti-inflammatory Study: study the HRBC membrane lysis and albumin denaturation	Respiratory infection	-	[54]
			ZP: −17.53 mV	In vitro release: 86.03 ± 0.06%				
			%EE: 84.36% ± 0.42%					
	Phytosome complexed with chitosan	Antisolvent precipitation	PS: 254.01 ± 0.05 nm	Drug release: sustained release				
			ZP: −13.11 mV	In vitro release: 88.93 ± 0.33%				
			%EE: 86.02% ± 0.72%					
*Moringa oleifera* leaf extract	Phytosome	Thin film (solvent evaporation)	PS: 198 ± 21 nm		Cytotoxicity assay: MTT assay	Wound dressing	Does not cause cytotoxicity at concentrations <1.5 mg/mL	[55]
			ZP: −28.30 ± 1.31 mV		Cell migration assay: NHDF cells			
*Lantana camara* extract	Phytosome	Solvent evaporation	%EE: 82.80%	In vitro drug release of 23% drug at 60 min	Dose: 20 mg/mL	Antibacterial and antifungal		[56]
*Murraya koenigii* extract	Phytosome	Antisolvent precipitation	PS: 236 nm	Sustain released:	Streptozotocin-nicotinamide induced diabetes model in male Wistar rats	Antidiabetic		[57]
			%EE: 75.1%	Release in 6 h:				
			ZP: −16.85 mV	Phytosome: 30%				
				Crude extract: 50%				

* Nanoparticle characteristics: particle size (PS); zeta potential (ZP); polydispersity index (PDI); encapsulation efficiency (EE).

**Table 3 nanomaterials-12-04073-t003:** Application of polymer-based delivery systems.

Bioactive Compound	Nanoparticle Type	Method of Preparation	Nanoparticle Characteristics *	Drug Release	Experiment/Model/Dose/Route	Application	Toxicity	Ref
*Jatropha pelargoniifolia* extract	Chitosan nanoparticles	Ionic gelation	PS: 185.5 nm,	JP-CSNPs: depend on pH of the medium	Antioxidant: DPPH	Antimicrobial and anticancer	-	[58]
			ZP: 44 mV,		Anticancer: In vitro cytotoxicity studies using A549			
			%EE: 78.5%	JP extract: showed controlled release	Human lung adenocarcinoma cells			
Piperin	Nanocapsule	Emulsion-diffusion	PS: 168.2 nm	Drug release: sustained release	In vitro (tested for growth inhibition study in axenic culture for 3 days)	Antitrypanosomal	Piperine-nanocapsule showed a safer index of safety against horse peripheral blood mononuclear cells (PBMC) compared to pure piperine.	[59]
			ZP: −20.3 mV		IC50 PNCs 5,04 uM; IC50 piperin 14,45 uM			
			PDI: 0.265					
*Argyreia pierreana* ethanolic crude extract (APEECE)	Mixed Micelles	Film dispersion	PS: 163 ± 10 nm		Type 2 diabetes induced rats using a high-fat diet (HFD) and low-dose (35 mg/kg) streptozotocin (STZ) injection	Antidiabetic and antihyperlipidemic		[60,61]
			PDI: 0.271 ± 0.07		dose APEECE & MDECE: 400 mg/kg			
*Matelea denticulata* ethanolic crude extract (MDECE)			PS: 145 ± 8 nm		Dose APEECE-Micelles & MDECE-Micelles: 200 mg/kg			
			PDI: 0.226 ± 0.08					
Curcumin	Chitosan/PEG blended PLGA nanoparticles	Emulsion solvent evaporation	PS: 264 nm		Apoptosis analysis: Annexin V assay by flow cytometer	Pancreatic cancer		[61]
			PDI: 0.181					
			ZP: 19.1 mV					
			%EE: 60%					

* Nanoparticle characteristics: particle size (PS); zeta potential (ZP); polydispersity index (PDI); encapsulation efficiency (EE).

### 3.1. Phytosomes Increasing the Activity of Phytochemical Compounds

Sinigrin is a glucosinolate found in the Brassicaceae family [62]. Mazumder et al. formulated sinigrin into a phytosome delivery system, intending to increase bioavailability and overcome the problem of the solubility of sinigrin, which was then assessed through its activity. It is known that sinigrin has wound-healing activity [49]. In a study conducted by Mazumder et al. (2016), the wound-healing activity of sinigrin was compared with the activity of the phytosome–sinigrin complex [62] Sinigrin-phytosome showed a significant wound-healing effect when compared to pure sinigrin. After 42 h, the phytosome–sinigrin complex healed 100% of the wounds, whereas pure sinigrin only showed 71% wound closure [49]. This proves that sinigrin-phytosome increases wound-healing activity.

Taxifolin, also known as dihydroquercetin, is a bioactive flavonoid that has an antioxidant effect [50]. Taxifolin can eliminate the excess of free radicals, enhancing immune functioning and reducing cancer cell formation in the human body [63]. From a literature study, it was shown that taxifolin has anticancer and antiproliferative effects on murine skin fibroblasts and human breast cancer cells [64]. However, taxifolins have limitations in their hydrophilicity and have a large molecular size, limiting their clinical application [65].

Kumar, et al. (2021) conducted a study in which a taxifolin ethyl acetate fraction derived from Cedrus deodara bark extract was loaded into a phytosome delivery system, with aims to evaluate the antioxidant and anticancer activity of taxifolin [50]. Phytosomes can increase the lipophilicity of the active compound so that they enhance the absorption of the active substance and increase its ability to cross biological membranes [66]. Based on the research of Kumar, et al. (2021), phytosomes showed high antioxidant activity with a lower IC50 value compared to the ethyl acetate and the standard. The results of the TB assay and MTT assay using the MCF7 cell line helped to conclude that the phytosome dilution (100 mg) showed significant anticancer activity compared to the ethyl acetate fraction [50].

Silybin is one of the main polyphenols that has been shown to exhibit many biological activities, including protecting the liver from oxidative stress and cancer formation as well as ongoing inflammatory processes [51]. However, this silybin has poor water solubility and low gastrointestinal absorption [67]. In order to overcome these limitations, Chi et al. (2020) conducted a study in which silybin was loaded into the delivery system of phytosome-nanosuspensions to enhance the hepatoprotective activity of silybin, which was tested in CCl_4_-induced model mice [51].

The induction using CCl_4_ resulted in an increase in the serum levels of ALT, AST, and AKP in the positive controls. Mice treated with silybin phytosome-nanosuspensions (SPc-NPs) showed significant reductions in ALT, AST, and AKP after CCl_4_ induction. In addition, histology of the liver sections from the mice treated with SPc-NPs was performed. The histology results showed clearly that SPc-NPs prevented hepatocyte necrosis, and also provided fewer fatty acid particles compared to the silybin-treated group. This is evidence of the enhanced hepatoprotective effect of Silybin phytosome-nanosuspensions [51].

Apigenin is a hydrophobic flavonoid compound that exhibits many biological activities, such as antioxidant, antimicrobial, anti-inflammatory, antiviral, antidiabetic, etc. [68,69,70,71]. Apigenin is reported to have excellent antioxidant and hepatoprotective properties. However, apigenin has limitations, namely, poor water solubility, fast metabolism, and low oral bioavailability, thus limiting its clinical application [72,73]. In order to overcome the limitations of apigenin, Telage et al. (2016) conducted a study where apigenin was loaded into a phytosome delivery system [52].

After being loaded into the phytosome delivery system, the solubility of apigenin-phytosome (APLC) in water increased compared to pure apigenin. In addition, the release of APLC was higher than that of pure apigenin, and the bioavailability of APLC increased compared to pure apigenin; this could be seen from the parameters Cmax, Tmax, and AUC. Apigenin activity as an antioxidant was tested by CCl_4_-induced hepatotoxicity models in rats; APLC showed significant antioxidant activity compared to free apigenin. This can be seen from the significant decrease in antioxidant activity indicators, such as glutathione, superoxide dismutase, catalase, and lipid peroxidase [52].

Quercetin is known to have poor absorption when given orally. In order to overcome this problem, El-Fattah et al. (2017) loaded quercetin into a phytosome delivery system to increase absorption and increase quercetin activity. Quercetin, as a phytoestrogen, is known to stimulate estrogen receptors (ERα and ERβ) [74]. In this study, quercetin was used for hormone replacement therapy [53].

Menopause symptoms that occur in women (aged 45–55 years) include an increase in food intake and body weight, metabolic dysfunction, bone density loss, diabetes, impaired muscle function, hyperlipidemia, psychological and mood changes, increased inflammatory markers, and oxidative stress, which can result in cell membrane lipid peroxidation and protein and DNA damage [75] Therefore, hormone replacement therapy has been the standard approach to relieve these symptoms.

The estrogenic activity of quercetin was observed at doses of 10 and 50 mg/kg/day using a mouse model that was ovariectomized for four weeks. These estrogenic activities include inflammation, oxidative stress, lipid profile, blood sugar levels, and bone and weight gain [53].

Treatment with quercetin-phytosome showed an increase in serum inorganic calcium phosphorus and glutathione content, as well as an improvement in the lipid profile. In addition, qurcetin-phytosome was found to decrease serum alkaline phosphatase, acid phosphatase, malondialdehyde, tumor necrosis factor-alpha, and blood sugar levels [53].

Quercetin-phytosome at a dose of 50 mg/kg showed better results than that of free quercetin. This is due to the better absorption of quercetin-phytosome. The results showed that antioxidant, anti-inflammatory, hypolipidemic, and other activities increased after quercetin was loaded into the phytosome delivery system [53].

Singh et al. (2018) carried out a gingerol formulation that was loaded into a phytosome delivery system, which was then complexed with chitosan to overcome the problem of respiratory tract infections [54]. Gingerol is a polyphenol derived from *Zingiber officinale,* which has many biological activities, one of which can protect cells against oxidative stress. In addition, gingerols can boost the immune system caused by free radicals, viruses, etc. However, these gingerols exhibit low bioavailability and water solubility profiles [76,77,78].

The results of research conducted by Singh et al. (2018) showed that gingerol-phytosome complexed with chitosan (GPLC) had better inhibitory activity than gingerol-phytosome (GP) against Gram-positive and Gram-negative organisms. This can be seen from the results of the in vitro tests on antioxidant, antimicrobial, and anti-inflammatory activities (Table 4, Table 5 and Table 6), as well as the in vivo tests on the hematological parameters (pharmacodynamic studies), where GP and GPLC showed effective antibacterial and anti-inflammatory activity against those bacterial and inflammatory organisms responsible for respiratory tract infections [54].

*Moringa oleifera* (MO) is known to have wound-healing activity, which has been tested both in vitro and in vivo [79,80] In addition, the aqueous extract of MO reduced the levels of inflammatory marker macrophages, including nitric oxide synthase (iNOS), tumor necrosis factor-α (TNF-α), and interleukin-1β (IL-1β) [81] Based on the results of the phytochemical screening, the aqueous extract of MO contains chlorogenic acid, gallic acid, kaempferol, quercetin, and vicenin-2, which are compounds based on in vitro studies and have anti-inflammatory properties [82]. In addition, these compounds can also increase wound-healing activity [79,80,83,84] However, this aqueous MO extract has limitations due to its large molecular size and high water solubility, thus inhibiting topical absorption and, thereby, limiting its ability to pass through lipid-rich biological membranes [85] Therefore, Lim et al. (2019) conducted a study in which an aqueous extract of MO was formulated in a phytosome delivery system (MPOCT). The results of the formulation development showed that the migration and proliferation rate of normal human dermal fibroblast cells (NHDF) was the highest compared to the control (MO water extract). In addition, MOPCT did not cause cytotoxicity at concentrations <1.5 mg/mL [55].

*Lantana camara* Linn. (*Verbenaceae*) is a flowering plant that contains saponins, alkaloids, proteins, carbohydrates, and glycosides, and contains a little amount of tannin [56,86]. Chime et al. (2020) formulated a methanol extract from *Lantana camara* leaves and used it in a phytosome delivery system (LCP) to observe the resulting antibacterial and antifungal activity. The results showed that LCP produced a larger diameter for the inhibition zone (IZD) compared to Lantana leaf methanol extract against *E. coli, L. ivanovii,* and *C. albicans*. The IZDs of *E. coli and L. ivanovii* were LC 10 mm, LCP 30 mm, LC 23 mm, and LCP 28 mm, respectively. This shows that LCP has better antibacterial and antifungal activity than LC [56].

*Murraya koenigii* (Linne.) Spreng belongs to the family *Rutaceae*, whose leaves are rich in carbazole alkaloids, coumarins, and acridine alkaloids [57]. *M. koenigii* exhibits many biological activities, one of which is an antidiabetic property [87]. However, *M. koenigii* has limitations in that its absorption through biological membranes is low, thus causing low bioavailability [57]. Therefore, Rani et al. (2022), conducted a formulation study by making phytosomes using a factorial design to overcome the limitations of *M. koenigii*. The phytosome delivery system not only enhances absorption by changing release characteristics but can also provide the sustained release of phytochemicals. From the results of these studies, it was shown that phytosome-*M. koenigii* causes a decrease in the serum glucose concentration of up to 42% at a lower dose compared to 39% for M. *koenigii* extract. This shows an increase in its therapeutic efficacy [57].

### 3.2. Polymeric Nanoparticle to Increase the Activity of Phytochemical Compounds

In a study conducted by Rani et al. (2020), the formulation of *piperine*-loaded nanocapsules (PNCs) was carried out to determine the resulting antitrypanosomal activity against *Trypanosoma evansi*, which causes trypanosomiasis [59]. *Piperine is an alkaloid compound and the main bioactive component in pepper. Piperine has been reported to exhibit antioxidant, antitumor, antiasthmatic, antipyretic, analgesic, anti-inflammatory, antibacterial, and antifungal activities. In addition, piperine has* been reported to induce toxic and apoptotic effects in protozoan parasites [88,89]. However, piperine has limitations, including low water solubility, fast metabolism, and undergoes systemic elimination [89,90]. This can be overcome by a nanotechnology-based delivery system; in this case, piperine is loaded into polymer nanocapsules using the emulsion-diffusion method [59].

Piperine loaded into a nanocapsule has a sustained release compared to pure piperine. The Piperine and PNCs were tested for growth inhibition studies in an axenic culture for three days. Piperine loaded into NCs (PNCs) showed significantly more antitrypanosomal activity with a 3.0-fold lower IC_50_ value than pure piperine (IC_50_ PNCs 5.04 M; IC_50_ piperine 14.45 M) [59]. Piperine nanoencapsulation, in the form of nanocapsules, showed a more significant antitrypanosomal effect than pure piperine. This is due to the increased absorption, bioavailability, and sustained release of piperine from the delivery system.

*Jatropha pelargoniifolia* (JP) is a medicinal plant rich in phenolics and flavonoids, which could be the major contributors to oxidative defense mechanisms in human cells. JP is widely used in traditional medicine because of its broad therapeutic activity [91]. However, JP has limitations, including poor solubility, bioavailability, and stability, and is sensitive to gastric acid pH so it cannot be administered in conventional dosage forms [92,93,94]. Alqahtani et al. (2021) conducted a study to overcome these limitations with nanotechnology, where JP-loaded chitosan nanoparticles (JP-CSNPs) were investigated for their antioxidant activity with antibacterial and anticancer potential [58].

The antioxidant activity of JP-CSNP was evaluated by a 2,2-diphenyl-1-picrylhydrazyl (DPPH) assay. The results showed that the antioxidant activity of JP-CSNP was slightly higher than that of the pure extract. The DPPH results showed that JP-CSNP had higher antimicrobial activity against Gram-positive bacteria with a 1.6-fold lower IC50 than the empty nanoparticles. The anticancer activity of the extracts of JP and JP-CSNP was tested using A549 adenocarcinoma human alveolar cells. The IC50 value of JP-CSNPs was two times lower than that of the JP extract [58]. Based on the results of the DPPH test and cytotoxic test on A549 human lung adenocarcinoma cells, it was shown that the *Jatropha*
*pelargoniifolia* (JP) contained in chitosan nanoparticles antimicrobial and anticancer potential, where its activity is slightly higher than the free extract.

Gudise et al. (2021) developed a mixed micelle nanoformulation (MM) delivery system with the active ingredients of *Argyreia pierreana* ethanol extract (APEECE) and *Matelea denticulata* ethanol extract (MDECE), which aimed to have dual pharmacological effects, namely as an antidiabetic and antihyperlipidemic in type 2 DM rats [60]. MM has the advantage that it can increase its solubility in water, has improved stability and pharmacokinetic properties, and can make the drug circulation time longer when compared to conventional micelles (made from single copolymers) [95].

Diabetes mellitus causes hyperglycemia and impaired glucose metabolism, causing an increase in lipid and lipoprotein levels and the production of free radicals [60]. Based on the results of phytochemical screening, both extracts contain flavonoids, phenols, and terpenoids, which are generally claimed to be useful for antidiabetic activity and antihyperlipidemic activity, according to previous reports [96,97,98,99].

The results of research by Gudise et al. (2021) showed that the antidiabetic, antihyperlipidemic, and antioxidant activities were significantly higher than the micellar nanoformulation. The MDECE-MM formulation had higher activity than APEECE-MM [60]. These results indicate that herbal formulations with nanotechnology (micelles) increase the activity of the herbal extracts. 

Curcumin is known to have biological activity against several diseases, such as neurological disorders, inflammatory diseases, diabetes, and various types of cancer [100]. Recent studies have demonstrated that curcumin can inhibit metastatic tumor spread in a pancreatic cancer xenograft model [101]. However, curcumin has limitations in terms of low water solubility, poor stability, and decreased bioavailability [102]. One strategy to increase the bioavailability and activity of curcumin is the development of a nano delivery system [103]. Therefore, Arya et al. (2018) conducted a study on *curcumin*-loaded chitosan/PEG-blended PLGA nanoparticles (CNP) as an ideal delivery system [60]. Based on the in vitro studio results, the CNP formulation showed strong cytotoxicity, enhanced antimigratory and antiinvasive properties, and induced apoptosis in metastatic pancreatic cancer compared to free *curcumin* [61].

## 4. Challenges of Phytochemical Formulations via Nanotechnology

*Taxol* is a natural alkaloid derived from the bark of the poisonous Pacific yew tree *(Taxus brevifolia). Taxol* is a potent natural anticancer agent that is able to reduce the growth rate of tumor cells by blocking cell replication during mitosis. Despite its outstanding anti-tumor properties, *taxol* is difficult to isolate, resulting in very low yields and high costs [104] The first formulation of *taxol* contains a cremophor EL (polyethoxylated castor oil), which functions to increase the solubility of *taxol* [105]. However, the cremophor was later found to be too toxic to living cells, including endothelial and epithelial cells [106]. Therefore, further research was conducted on *taxols* that lead to nanoparticle involvement. *Taxol* (Paclitaxel) has been approved by the FDA (brand name: Abraxane) for the treatment of breast cancer in the formulation of paclitaxel as albumin-bound nanoparticles [107]. It is known that the improvisation of *taxol* using albumin-bound nanotechnology is less toxic than chremophor-*taxol* [108]. The paclitaxel-albumin nanoparticle formulation has a modified pharmacokinetic profile so that it has higher solubility and better tissue distribution. To date, Abraxane is used for the treatment of breast cancer, nonsmall cell lung cancer, and pancreatic cancer.

The extract of *G. biloba* (GbE) was reported to exhibit several activities, including scavenging radical, autooxidation, antitumor, and protective effects in the central nervous system [109]. However, for the oral administration of GbE, low bioavailability (10%) and a short half-life (2.1 h) have caused problems regarding bioavailability [110]. The use of delivery systems, such as niosomes, generally increases drug diffusion through biological membranes and provides protection (to the drug) from enzymatic degradation, thereby increasing drug bioavailability. Jin et al. (2013) carried out the encapsulation of GbE in a niosome delivery system. Stability studies revealed that drug entrapment efficiency for the GbE niosomes at 4 °C and 25 °C after three months of storage has good stability [109]. The addition of cholesterol has been reported to exhibit stabilization effects on the bilayer by preventing the leakage of the drug components and inhibiting the permeation of the solutes enclosed in the aqueous core of the niosomes [111] In vitro release studies showed that GbE niosomes could prolong the release of flavonoid glycosides in a phosphate buffer (pH 6.8) for up to 48 h. In vivo distribution studies showed that the content of flavonoid glycosides in the heart, lungs, kidneys, brain, and blood of rats treated with the GbE niosome carrier system was greater than that in rats given oral GbE tablets [109].

Nanotechnology-based delivery systems for active herbal ingredients offer many advantages, including increased solubility, bioavailability, pharmacological activity, stability (of active ingredients), protection (over the active ingredients) from chemical and physical degradation, and the reduced dose required. The potential of developing herbal medicine as a promising alternative medicine cannot be denied. However, formulations using nanotechnology face their own challenges, such as high production costs, the difficulty of scaling up the process, and the lack of data regarding the safety and toxicity of herbal formulations using nanotechnology; therefore, the potential hazards associated with nanotechnology-based drug delivery systems must be considered. Another challenge for nanoparticles is the stability of the nanoparticles themselves. The issue of the ability of nanoparticles to retain drugs is increasing because some nanoparticles have been shown to leak after contact with blood components [112].

Nanosystems in phytochemicals are significantly more expensive to manufacture, leading to higher selling prices. In European countries, drugs are selected and funded from public sources based on rational selection, where high production costs can be a serious obstacle. In these countries, high-priced nanomedicines have a very low chance of reaching the market, as well as the patients [113].

## 5. Conclusions

The application of nanotechnology-based delivery systems for phytochemical constituents plays an important role in public healthcare worldwide. The use of herbal medicines is growing worldwide, but their disadvantages of low solubility, bioavailability, and pharmacological activity, as well as being physically and chemically unstable and easily degraded, limit their clinical application as medicines. Therefore, the development of herbal medicines with nanotechnology-based delivery systems might be an alternative strategy for increasing their pharmacological activity. However, the development of these nanotechnology-based delivery systems still needs to be reviewed further, especially regarding safety and toxicity profiles, so that their safety and effectiveness for curing various types of diseases can be ensured.

## Figures and Tables

**Figure 1 nanomaterials-12-04073-f001:**
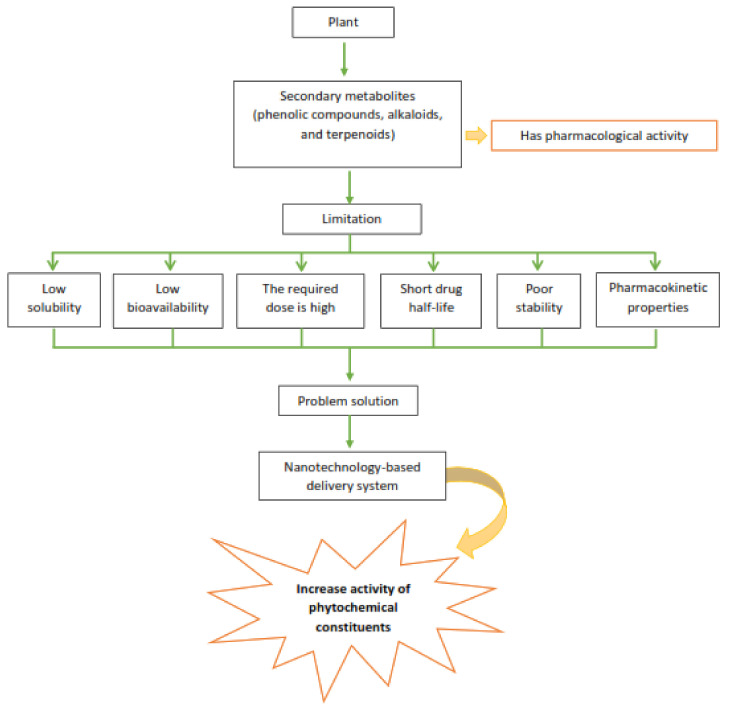
Schematic showing the deficiencies of the phytochemicals that limit their clinical application. Nanotechnology-based delivery system methods can overcome these limitations, mostly by increasing their bioavailability and absorption, thereby increasing their activity [20,21,22,23].

**Figure 2 nanomaterials-12-04073-f002:**
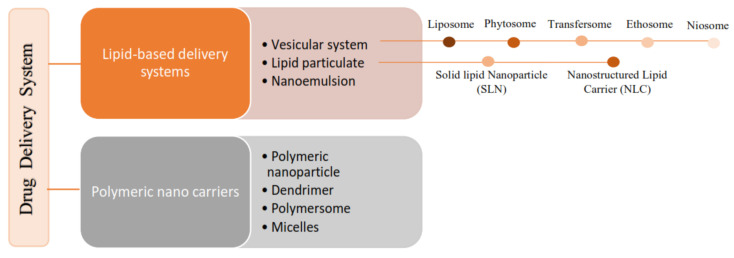
The nanotechnology-based delivery system for phytochemical constituents [19,26].

**Figure 3 nanomaterials-12-04073-f003:**
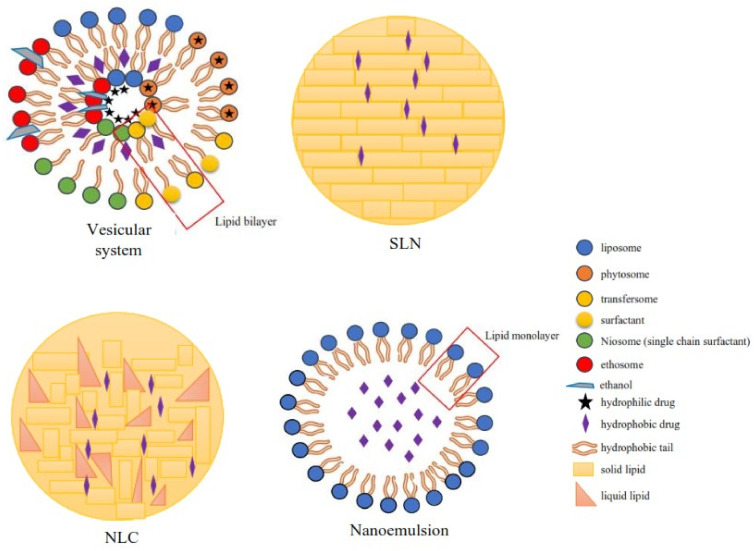
Classification of the lipid-based delivery system.

**Figure 4 nanomaterials-12-04073-f004:**
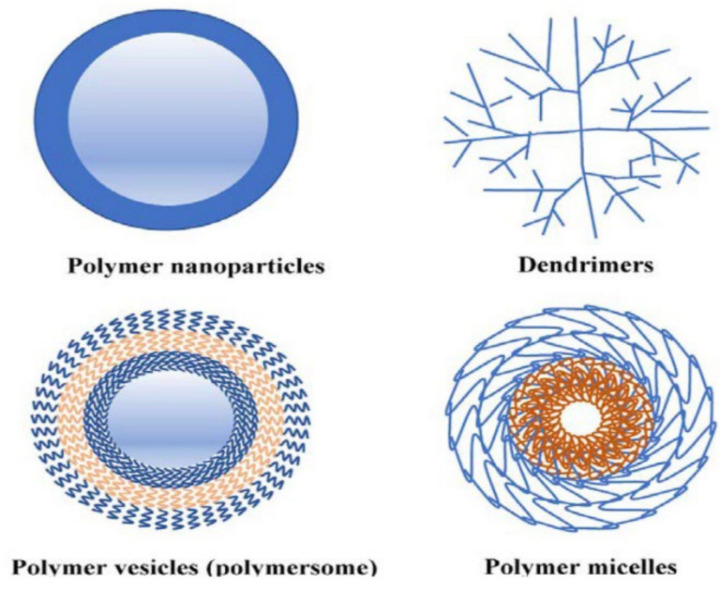
The classification of polymeric nano carriers (adapted from [44]).

**Table 1 nanomaterials-12-04073-t001:** Comparative of Lipid-based Nanocarrier [45,46].

Lipid-Based Nanocarrier	Composition	Structure	Application	Administration
Liposome	Phospholipid and cholesterol	Rigid	Drug (synthetic and naturally derived) and gene therapy	Oral, parenteral, topical, and transdermal
Phytosome	Phospholipid and polyphenolic phytoconstituents	Rigid	Phyto delivery	Oral, parenteral, topical, and transdermal
Transfersome	Phospholipid and surfactant	Ultra-deformable	Skin delivery	Topical and transdermal
Niosome	Nonionic surfactant and cholesterol	Rigid	Drug (synthetic and naturally derived) delivery and cosmetics	Oral, parenteral, topical, and transdermal
Ethosome	Phospholipid, alcohol, polyglycol, and water	Elasticity	Skin delivery	Topical and transdermal
SLN	Solid lipid	Ordered	Drug (synthetic and naturally derived), gene therapy, and cosmetics	Oral, parenteral, topical, and transdermal
NLC	Solid lipid and liquid lipid	Less ordered	Drug (synthetic and naturally derived), gene therapy, and cosmetics	Oral, parenteral, topical, and transdermal

**Table 4 nanomaterials-12-04073-t004:** In vitro antioxidant study (adapted from Refs. [54,56]).

Method	IC_50_		
	Gingerol	GP	GPLC
DPPH assay	57.74 µg/mL	46.23 µg/mL	17.70 µg/mL
H_2_O_2_ assay	53.52 µg/mL	40.48 µg/mL	19.46 µg/mL

**Table 5 nanomaterials-12-04073-t005:** In vitro microbial study [54].

Organism	MIC		
	Gingerol	GP	GPLC
Gram positive *(S. aureus)*	400 ± 0.23 µg/mL	200 ± 0.06 µg/mL	100 ± 0.07 µg/mL
Gram Negative *(E. coli)*	400 ± 0.12 µg/mL	200 ± 0.79 µg/ml	100 ± 0.08 µg/mL

**Table 6 nanomaterials-12-04073-t006:** In vitro anti-inflammatory effect [54].

Organism	IC_50_		
	Gingerol	GP	GPLC
Against HRBC membrane lysis	74.68 µg/mL	70.86 µg/mL	59.84 µg/mL
Denaturation of protein	67.03 µg/mL	64.54 µg/mL	61.88 µg/mL

## Data Availability

Not available.

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
