# Peer review of "Improved Activity of Herbal Medicines through Nanotechnology"

_nanomaterials, 2022, doi:10.3390/nano12224073_

Round 1

Reviewer 1 Report

This review article aims to provide an overview of the latest advances in the development of nanotechnology-based herbal drug formulations to increase their activity as well as the challenges of the nanotechnology-based delivery system. This is a topic with great application prospects.

 However, the following three points need to be improved:

1.  The title of the article is to Improved activity of phytochemical by nanotechnology. but the full text mainly describes how to improve the activity of herbal medicine by. Nanotechnology and its importance. The title of the article is changed to Improve the activity of herbal medicine through nanotechnology, which is more consistent with the paper.

2. Please add the word "herbal medicine" to the keywords in the abstract.

3. Please add at least 3 references in recent three years.

Reviewer 2 Report

Dear Authors,

The manuscript „Improved Activity of Phytochemical by Nanotechnology” submitted for consideration to Nanomaterials presents an interesting and actual subject, referring to various research areas including phytochemistry, pharmaceutical technology, nanotechnology etc.  Even though the topic is worth describing, the presented manuscript displays several flaws which make it unsuitable for publication in its current form. For my remarks and questions, please see the attached file.

Reviewer 3 Report

Dear authors,

this review article aims to provide an overview of the latest advances in the development of nanotechnology-based herbal drug formulations to increase their activity as well as the challenges of the nanotechnology-based delivery system. It is well written however I would add a paragraph or some examples of encapsulation of phytochemicals within liposomes which are being very important molecules for drug delivery and are not never mentioned in this review.

Best Regards

Round 2

Reviewer 2 Report

The manuscript has been thoroughly checked and most of the issues indicated in the previous revision were addressed correctly. The previously mentioned parts have been improved. However, I would suggest some minor corrections. Please see the list below:

1.     Introduction, lines 50-51: perhaps it would be better to say that some compounds are sensitive to acidic conditions. Please correct the typographical errors.

2.     Lines 55-56: please correct to: Alzheimer’s disease, Parkinson’s disease

3.     The manuscript still contains some typographical errors. Please check the text and the new figures thoroughly.

4.     Lines 179-180: please reformulate the sentence, it seems to be incorrect.

5.     Tables 4 - 6 are not mentioned in the text in any way.

6.      Line 395: perhaps it should be Cremophor instead of chromophore. In fact, there are at least two compounds frequently used as solubilizers. Please specify which one was described here.
